# Clusters of diet, physical activity, screen-time and sleep among adolescents and associations with 3-year change in indicators of adiposity

**Noura Alosaimi**[1]*, **Lauren B. Sherar**[1], **Paula Griffiths**[1], **Mark Hamer**[2], **Natalie Pearson**[1]

**1** School of Sport, Exercise and Health Sciences, Loughborough University, Loughborough, Leicestershire, United Kingdom, **2** Faculty of Medical Sciences, University College London, London, United Kingdom

* n.n.alosaimi@lboro.ac.uk

**Data Availability Statement:** Our study uses data obtained from the UK Data Service under the terms of their End User Licence (EUL), which includes restrictions on data sharing. According to the EUL,

## Abstract

### Background

Clusters of health behaviours could impact changes in adiposity among adolescents over time. This study examines the clustering of screen time, physical activity, dietary behaviours and sleep, and the associations with 3-year changes in indicators of adiposity.

### Methods

Data from the UK's Millennium Cohort Study were utilised when participants were aged 14 and 17 years respectively. At age 14, demographics, screen time, dietary behaviours and sleep duration were measured via self-report, and physical activity using wrist worn accelerometers. Height, weight, and percent body fat were measured at age 14 and 17 years. Behavioural clusters were determined using k-means clustering analysis, and associations with change in indicators of adiposity between age 14 and 17 years were examined using multivariate regression models.

### Results

Three clusters were identified at age 14, a 'healthy cluster', a 'mixed cluster', and an 'unhealthy cluster' in the analytical sample of 3,065 participants (52.5% girls). The 'unhealthy' cluster was the most prevalent cluster among boys (53%), while the 'healthy cluster' was most prevalent among girls (55.9%). Adolescents in healthy clusters had a lower BMI z-score and percent body fat at age 14 compared to those in the unhealthy and mixed clusters, and maintained lower scores at age 17. Boys in the mixed and unhealthy clusters at 14 years had a lower change in BMI z-score between 14 and 17 compared to boys in the healthy cluster.

### Conclusion

Adolescents in the healthy cluster had lower BMI z-scores and percent body fat at age 14 years than those in the unhealthy cluster, and they maintained this lower level at age 17.

we are not permitted to share the data directly with third parties, as this is restricted to authorized users who have registered and agreed to the EUL terms, which include any relevant special conditions. The UK Data Service imposes these restrictions due to the sensitive nature of the data, which includes potentially identifiable information. Requests for access to the data should be directed to the UK Data Service, which manages these data under strict access conditions to protect participant confidentiality. Contact Information for Data Access Requests: UK Data Service Website: https://ukdataservice.ac.uk For this study, we made use of the MCS1, MCS2, MCS6, and MCS7 surveys.

**Funding:** The author(s) received no specific funding for this work.

**Competing interests:** The authors have declared that no competing interests exist.

Given the upward trend in BMI during this period, this maintenance could be interpreted as a positive outcome. Further prospective research is needed to better understand these associations as well as research examining the stability of cluster membership over time.

## Background

Diet, physical activity, sedentary behaviour and sleep are important modifiable contributors to obesity and its associated comorbidities [1]. Recent evidence suggests that only 18% of adolescents in the UK consume the recommended five portions of fruits and vegetables daily [2], and less than 10% meet the recommendations for sleep, screen-time, and physical activity simultaneously [3]. Health behaviours established during adolescence often become deeply ingrained and can persist into adulthood, shaping long-term health behaviours [4]. Therefore, adolescence appears to be a critical period warranting public health efforts to improve health behaviours.

Evidence suggests that poor dietary behaviours, low levels of physical activity, high levels of sedentary behaviours and poor sleep tend to cluster in young people [5, 6]. Each of these health behaviours individually is known to contribute to adiposity in youth [7], but together may have synergistic effects leading to poor health [5, 8]. Methods that categorise young people into distinct clusters based on their lifestyle behaviours are becoming increasingly popular for investigating the complex relationships between these behaviours and health outcomes. The prevalence of clusters of unfavourable health behaviours has been shown to increase from 29% in children aged 2–5 years to 73.9% among 16–19-year-olds, with the combination of excessive screen time and poor dietary habits emerging as the most frequent, increasing in prevalence from 14.4% in young children to 45.3% in older adolescents [9]. Whereas the prevalence of clusters of favourable health behaviours (i.e. low sedentary behaviours, high moderate to vigorous physical activity (MVPA) and diet quality) was only 18% in a European study of adolescents [10].

Most recent studies examining the associations between clusters of health behaviours and adiposity have been cross-sectional [11–13]. For example, clusters of unfavourable health behaviours have been associated cross-sectionally with increased body mass index (BMI) in adolescent girls and with higher odds of increased waist circumference and percent body fat in adolescent boys [14]. A recent systematic review synthesised the evidence on the prevalence of clustering of physical activity, sedentary behaviours, and diet among children, adolescents, and young adults and their associations with physical and mental health indicators, and found that individuals in both unhealthy and mixed clusters had higher adiposity [15]. The review also highlighted a scarcity of longitudinal studies and few studies that had included sleep in the analysis of clustering of behaviours. Furthermore, there was a noticeable gap in research examining clustering of health behaviours and associated health outcomes in groups of older adolescents [16]. Although current research on the clustering of such lifestyle behaviours has increased [17–21], little is known about in whom and how health behaviours cluster among adolescents, and whether clusters of health behaviours are associated with change in weight and percent body fat during later adolescence. During this critical period of rapid physical growth and development, the lifestyle behaviours established can have long-lasting effects on health outcomes in adulthood [4]. Therefore, interventions targeting healthy behaviours during later adolescence can be particularly impactful in promoting long-term health and well-being.

The current study is original because it examines the clustering of health behaviours among adolescents, and the associations between clusters and change in adiposity in a large sample of adolescents from the UK. This is significant as it reveals how different health behaviours affect adiposity, informing targeted interventions to reduce adolescent obesity. The aims of this study were to (1) identify clusters of lifestyle behaviours based on fruits and vegetables and sugar sweetened beverages (SSB) intake, MVPA, screen-time and sleep duration; (2) examine the associations between health behaviour clusters and change in (a) body mass index (BMI) z-score, (b) weight status, (c) percent body fat.

## Methods

### Sample and data collection

This study uses data from Millennium Cohort Study (MCS) [22]. The MCS is a nationally representative longitudinal cohort study of 19,243 babies born around the millennium (Sep 2000 –Jan 2002) in the four home countries of the United Kingdom. The MCS was designed to be multidisciplinary, examining a wide range of factors that influence the health and well-being of children and their families. Information from the main carers (mainly mothers) was first collected (i.e. weekly household income, highest maternal academic qualification level, and child's birthweight and gender) when babies were around 9 months of age and they have since been followed up at age 3, 5, 7, 11, 14, 17, and 23 years. The present study used data collected in 2015/2016 from sweep MCS6 (when children were aged 14 years: n = 11,872), primarily due to the availability of comprehensive measures for this research objective, and data collected in 2018/2019 from sweep MCS7 when children were aged 17 years (n = 10,757).

### Measures

All measures of health behaviours were assessed at age 14 years. Indicators of adiposity were assessed at age 14 and 17 years. This is a secondary data analysis, and all measures (i.e. questionnaires, protocols for device use) were developed by the original cohort team.

**Physical activity.** Physical activity was assessed using wrist worn triaxial GENEActiv accelerometers [23] (Activinsights Ltd, Kimbolton, UK). Full details on the accelerometer data collection protocol have been described elsewhere [23]. In brief, participants were instructed to wear the accelerometer on their non-dominant wrist for two specified full days, one weekday and one weekend day (randomly selected at the time of the interviews), between January 2015 and March 2016. Data were downloaded using GENEActiv software and processed using the GGIR package in R, which includes autocalibration and non-wear detection functions [24]. The data were collected in 5-second epochs and included in the present analyses if participants had $\geq$ 10 hours of valid wear for both days. Euclidean Norm Minus One (ENMO), a measure of mean acceleration over a 24-h period, was used to estimate the overall physical activity [23]. Duration of MVPA was calculated as the time spent with 80% bout criterion of $\geq$ 100 ENMO for at least 1 minute [25]. Adolescents were classified into two groups: those who achieved a daily average of 60 minutes of MVPA, and those who did not [26].

**Screen-time.** To examine time spent on screen-based behaviours at age 14 years (MCS6 survey), adolescents self-reported the number of hours per weekday spent (i) watching tv / videos / computers, (ii) playing electronic games, and (iii) on social networking sites. Adolescents responded to an 8-point Likert scale ranging from 'none' to '7 or more hours a day'. For the present analyses, each screen-time variable was categorised as < 2 hours and > 2 hours [27].

**Dietary behaviours.** Participants reported how often they ate at least 2 portions of fruit per day, 2 portions of vegetables per day, and how often they drank SSB during a week (i.e., once a day, 3–6 days a week, etc). For the present analyses, fruit and vegetable consumption

respectively, were dichotomised into 'Never/Some days' and 'Every day'. SSB consumption was dichotomised into 'more than 3 times per week' and 'less than 3 times per week' [28].

**Sleep.** Participants were asked about sleep, and selected sleep-onset time for both school and non-school days (i.e. weekends) from: "Before 9pm"/ "9–9:59pm"/ "10–10:59pm"/ "11–11:59pm"/ "After midnight", and on wake-up wake-up time for school days and non-school days from: "Before 6am"/ "6–6:59am"/ "7–7:59am"/ "8–8:59am"/ "After 9am" and "Before 8am"/ "8–8:59am"/ "9–9:59am"/ "10–10:59am"/ "11–11:59am"/ "After midday", respectively. Average sleep onset times were calculated using a weekday-to-weekend weighting ratio of 5:2 and grouped into four categories: Before 10pm (reference group), 10–10:59pm, 11–11:59pm, and after midnight. Sleep onset times were combined with usual wake-up times on both school days and non-school days to calculate average sleep durations. Sleep duration was calculated as the time between the category mid-points for sleep onset and wake time. These durations were weighted in a ratio of 5:2 for weekdays to weekends and were categorised into two groups: < 9 hours and > 9 hours.

**Adiposity.** Data on three indicators of adiposity (BMI, weight status and percent body fat) were collected during home visits conducted by trained researchers at ages 14 and 17 years. Height was measured using Leicester Height Measure Stadiometers (Seca, Birmingham, UK) to the nearest 0.1 cm. Weight and percentage body fat were assessed using calibrated Tanita BF-522W scales (Tanita UK, Yiewsley, Middlesex, UK). For ages 14 and 17, BMI was calculated by dividing weight in kilograms by squared height in metres and was used to derive BMI z-scores using UK 1990 growth centiles [29]. Weight status was represented as a binary variable (normal/under-weight, overweight or obese), based on age- and sex-specific criteria provided by the International Obesity Task Force [30].

## Potential confounding variables

Based on previous literature [31], potential confounding variables were considered from self-report data when adolescents were age 14 years. Ethnic group (white, mixed, Indian, Pakistani and Bangladeshi, Black/Black British, and other ethnic group), which was reclassified as (i) white and (ii) other. Parents reported their weekly household income (Organisation for Economic Co-operation and Development [OECD] equivalized income quintiles), highest maternal academic qualification level (degree plus, diploma, A levels, General Certificate of Secondary Education [GCSE] grade A to C, GCSE D to G, overseas only, and none), which was reclassified as (i) low, (ii) high, and (iii) Other qualifications, and adolescents' birthweight and gender during data collection at waves one and two (MCS1–MCS2). Adolescents reported their pubertal status, including growth spurt, body hair, and skin changes for both boys and girls, and additional factors such as voice breaking and facial hair for boys, and breast development and menarche for girls, at the age of 14 (MCS6) using a self-administered rating scale for pubertal development [32]. The following puberty categories were derived: prepubertal, early pubertal, midpubertal, late pubertal and postpubertal, which were dichotomised into (i) pre/early puberty, including 'prepubertal,' 'early pubertal,' and 'midpubertal,' and (ii) post/late pubertal, including 'late pubertal' and 'postpubertal' for the current analyses.

## Ethics

Ethical approval has been granted for all MCS surveys via the National Health Service Research Ethics Committee system [33]. This ensure that all data is collected in line with ethical guidelines, including informed consent, confidentiality, and participant welfare. Written consent for data collection was obtained from parents during the early waves when cohort members were under the age of 16, and as participants grew older, their own consent was required for

continued involvement [33]. Also, ethical issues extend to secondary data analysis, where researchers must adhere to guidelines for the appropriate and responsible use of data [33]. The MCS data can be obtained from the UK Data Service. After user and project registration with the UK Data Service, access to the MCS data was approved, and the datasets from sweeps 6 and 7 were downloaded onto an encrypted personal computer in anonymised STATA file format on November 12, 2020. Since the downloaded data were publicly available and fully anonymised, containing no personally identifiable information, no additional ethical approval was required for the current study [34].

## Statistical analyses

Data management and analyses were conducted using Stata statistical software, version 18 (Stata, College Station, TX). A sensitivity analysis showed a significant difference in one key health outcome between participants with complete and missing data, supporting the use of a complete-case dataset for accuracy and reliability. Therefore, only participants with complete data for all variables of interest at ages 14 and 17 were included. Descriptive analysis was performed to compare demographic characteristics between the included and excluded populations using appropriate statistical methods (S1 Table). Pearson's Chi-square test and independent samples t-tests were used to examine differences in baseline characteristics for categorical and continuous variables, respectively. A p-value $< 0.05$ was considered statistically significant.

Joint Correspondence Analysis (JCA) was used to explore patterns in categorical variables, involving four exposure measures (physical activity, screen-time, dietary behaviours, and sleep) with eight variables (MVPA, TV viewing, video game playing, social networking, sleep duration, and consumption of fruits, vegetables, and SSB). More details on the analyses used have been described elsewhere [35]. In brief, JCA was first conducted to reduce data dimensionality and to examine patterns and relationships in categorical variables. Then, K-means cluster analysis was utilised to create clusters of behaviours and was conducted using the coordinates obtained from the JCA. The algorithm was run with a predetermined number of clusters, k = 3, based on theoretical considerations and inspection of within-cluster sum of squares from the JCA. The resulting k-means clustering solution produced 3 distinct nonoverlapping clusters to group similar observations while maximising differences. The final number of clusters was determined and cross checked based on both JCA and k-means and their interpretability and distinctiveness of the cluster solution. The clusters' homogeneity was determined based on the sum of squared distances. Chi-square tests examined associations with sociodemographic variables.

Linear and logistic regression models, stratified by gender, were used to analyse the longitudinal associations between identified clusters at age 14 (baseline) and (1) BMI z-scores, (2) weight status categories, and (3) percent body fat at age 17 years, adjusted for maternal education, ethnicity, weekly family income, birthweight, puberty, baseline BMI z-score, baseline weight status, and baseline percent body fat, respectively.

## Results

### Participant characteristics

Complete data were available for 3,065 participants (52.5% girls), representing 16% of the original sample (see S1 Fig). Participants included in the analyses represented all UK countries with 30% of participants from England, 37% from Wales, 42% from Northern Ireland, and 44% from Scotland. The participants excluded from the analysis differed from those included in the analyses in terms of ethnicity and socioeconomic status (S1 Table).

The descriptive characteristics of the analytical sample at age 14 are presented in Table 1. The majority of boys (84.8%) and girls (84.6%) were white British, just over half of participants (53.8%) had mothers with a higher level of education. There were no significant differences in ethnicity, OECD weekly family income, and maternal education between boys and girls.

### JCA and k-means cluster analysis

The JCA results revealed two dimensions that explained 92.7% of the total variance within the dataset (i.e. the extent to which the two dimensions explain the variability among the variables included in the analysis). The scatter plot (Fig 1) displays the positions of the observations and the relationships between the physical activity, dietary, screen-time and sleep variables. K-means clustering solution produced three distinct clusters (Fig 2), which were termed healthy cluster, mixed cluster, and unhealthy cluster.

The characteristics of each of the three clusters are described in Table 2. For both boys and girls, Cluster 1 (healthy cluster) was characterised by a higher proportion of adolescents who met the physical activity guidelines, watched screens for less than 2 hours per day, had a sleep duration of more than 9 hours, consumed SSB less than 3 times a week, and consumed 2 portions of fruits and vegetables daily. Cluster 2 (mixed cluster) was characterised by values of health behaviours and health outcomes that fall between those observed in the healthy and unhealthy clusters. Cluster 3 (unhealthy cluster) was characterised by a higher proportion of adolescents who did not met the physical activity guidelines, never/some days ate 2 portions of fruits and vegetables, consumed SSB more than 3 times a week, watched screens for more than 2 hours, and had sleep durations of less than 9 hours.

At age 14, a higher proportion of boys were in the unhealthy cluster (53.0%) compared to girls, and a higher proportion of girls in the healthy cluster (55.9%) compared to boys. For both boys and girls, a higher proportion of adolescents with mothers who had higher educational levels were in the healthy clusters compared to those whose mothers had lower or 'other' levels of education (69.6% for boys, and 65.4% for girls). A higher proportion of adolescents with mothers with lower educational levels were in the unhealthy cluster compared to those whose mothers who had higher or 'other' levels of education (48.9% for boys, and 50.1% for girls).

Table 3 (unadjusted models) and Table 4 (adjusted models) represent the longitudinal associations between clusters at age 14 and changes in BMI z-score, weight status, and percent body fat between ages 14 and 17. Change in BMI z-scores were significantly lower among boys in the mixed and unhealthy clusters compared to those in the healthy cluster ($\beta$ = - 0.08; 95% CI = (-0.146, - 0.005)), $\beta$ = - 0.08; 95% CI = (- 0.158, - 0.001)), respectively). No significant changes were observed for weight status and percent body fat for both sexes.

### Discussion

This study identified and described the clustering of screen time, physical activity, dietary behaviours and sleep among adolescents and examined the associations with 3-year changes in indicators of adiposity. This study presents a novel approach by examining adolescence, a critical period of life, and exploring the associations between clusters of health behaviours and changes in weight over a 3-year period in a UK sample, with the inclusion of sleep patterns within the analysis of behaviour clustering. The present study identified three clusters, a 'healthy cluster,' 'mixed cluster,' and 'unhealthy cluster,' with the unhealthy cluster being the most prevalent among boys and the healthy cluster being the most prevalent among girls. We found that boys in the mixed and unhealthy clusters at age 14 years had a significantly lower

**Table 1. Participant characteristics and outcomes at ages 14 and 17 years.**

| | Girls N = 1,608 (52.46%) | Boys N = 1,457 (47.54%) | Total (N = 3,065) | *P* value |
|---|---|---|---|---|
| **Demographic characteristics** | | | | |
| **Ethnicity (n (%))** | | | | |
| White | 1,361 (84.64%) | 1,235 (84.76%) | 2,596 (84.70%) | 0.924 |
| Other | 247 (15.36%) | 222 (15.24%) | 469 (15.30%) | |
| **OECD weekly family income (GBP, mean (SD))** | 446.37 (176.62) | 457.27 (170.45) | 451.55 (173.77) | 0.0830 |
| **NVQ Highest Level (all sweeps, n (%))** | | | | |
| Low | 675 (41.98%) | 580 (39.81%) | 1,255 (40.95%) | 0.066 |
| High | 838 (52.11%) | 811 (55.66%) | 1,649 (53.80%) | |
| Other qualifications | 95 (5.91%) | 66 (4.53%) | 161 (5.25%) | |
| **Birthweight (kg, mean (SD))** | 3.32 (0.56) | 3.45 (0.58) | 3.38 (0.57) | < 0.001 |
| **Puberty (n (%))** | | | | |
| Pre/early puberty | 167 (10.43%) | 986 (67.86%) | 1,153 (37.75%) | < 0.001 |
| Post/late pubertal | 1,434 (89.57%) | 467 (32.14%) | 1,901 (62.25%) | |
| **Physical activity, sedentary behaviours, and sleep at age 14 years** | | | | |
| MVPA (median mins/day (IQR)) | 46.17 (29.88–70.29) | 59.96 (36.92–95.00) | 52.25 (32.31–81.48) | < 0.001 |
| Total accelerometer wear time (median hrs/weekend and weekday combined (IQR)) | 47 (42.5–48) | 46.25 (41.5–48) | 47 (42–48) | 0.0050 |
| **Screen time (median mins/weekday (IQR))** | | | | |
| Watching TV / videos / computer | 150 (90–240) | 150 (90–240) | 150 (90–240) | 0.0039 |
| Playing electronic games | 15 (0–90) | 150 (90–240) | 90 (15–150) | < 0.001 |
| Social networking sites | 150 (45–240) | 90 (15–150) | 90 (45–240) | < 0.001 |
| **Sleep duration (hrs /day, n (%))** | | | | |
| < 9 | 705 (43.84%) | 638 (43.79%) | 1,343 (43.82%) | 0.976 |
| > 9 | 903 (56.16%) | 819 (56.21%) | 1,722 (56.18%) | |
| **Dietary intake** | | | | |
| **Sweetened beverages (frequency/week, n (%))** | | | | |
| < 3 times a week | 1,049 (65.24%) | 828 (56.83%) | 1,877 (61.24%) | < 0.001 |
| > 3 times a week | 559 (34.76%) | 629 (43.17%) | 1,188 (38.76%) | |
| **At least 2 portions of fruit per day (n (%))** | | | | |
| Never / some days | 1,070 (66.54%) | 962 (66.03%) | 2,032 (66.30%) | 0.763 |
| Every day | 538 (33.46%) | 495 (33.97%) | 1,033 (33.70%) | |
| **At least 2 portions of vegetables per day (n (%))** | | | | |
| Never / some days | 934 (58.08%) | 879 (60.33%) | 1,813 (59.15%) | 0.207 |
| Every day | 674 (41.92%) | 578 (39.67%) | 1,252 (40.85%) | |
| **Outcomes (mean (SD))** | | | | |

(*Continued*)

**Table 1.** (Continued)

|  | Girls<br>N = 1,608<br>(52.46%) | Boys<br>N = 1,457<br>(47.54%) | Total<br>(N = 3,065) | *P* value |
|---|---|---|---|---|
| **Body fat % at 14** | 26.70 (6.77) | 16.27 (7.62) | 21.74 (8.87) | < 0.001 |
| **Body fat % at 17** | 27.93 (8.08) | 15.60 (7.93) | 22.07 (10.09) | < 0.001 |
| **BMI z-score at 14** | 0.64 (1.14) | 0.55 (1.15) | 0.60 (1.15) | 0.0202 |
| **BMI z-score at 17** | 1.12 (1.08) | 1.21 (1.05) | 1.16 (1.07) | 0.0147 |
| **BMI category 14 (n (%))** |  |  |  |  |
| Normal/Under-weight | 1,217<br>(75.68%) | 1,138<br>(78.11%) | 2,355<br>(76.84%) | 0.272 |
| Overweight | 288<br>(17.91%) | 238<br>(16.33%) | 526<br>(17.16%) |  |
| Obese | 103<br>(6.41%) | 81<br>(5.56%) | 184<br>(6.00%) |  |
| **BMI category 17 (n (%))** |  |  |  |  |
| Normal/Under-weight | 1,129<br>(70.21%) | 1,090<br>(74.81%) | 2,219<br>(72.40%) | 0.017 |
| Overweight | 316<br>(19.65%) | 244<br>(16.75%) | 560<br>(18.27%) |  |
| Obese | 163<br>(10.14%) | 123<br>(8.44%) | 286<br>(9.33%) |  |

Abbreviations: OECD, organisation for economic co-operation and development; GBP, great british pound; SD, standard deviation; NVQ, national vocational qualification; kg, kilogram; MVPA, moderate to vigorous physical activity; mins, minutes; IQR, interquartile range; hrs, hours; BMI, body mass index.

change in BMI z-score compared to those in the healthy cluster. No other associations were identified between clusters of health behaviours and changes in indicators of adiposity.

The present study found that the prevalence of adolescents in the healthy clusters at 14 years of age was around 30%, while the prevalence of adolescents in the unhealthy clusters was just below 30% [11, 15]. In support of this finding, previous systematic reviews show that the majority of adolescents are categorised into mixed clusters, where one or more healthy behaviour(s) coexists with one or more unhealthy behaviour(s), highlighting the intricate nature of adolescents' health behaviours. The present study also revealed that adolescents of mothers with higher education were more likely found in the healthy cluster and adolescents of mothers with lower education were more likely found in the unhealthy cluster; an observation also consistently observed in previous research [11, 15, 36]. In contrast to a recent review [15], we observed a higher prevalence of girls in the healthy cluster, while boys were predominantly in the unhealthy cluster. This maybe attributed, in part, to variations in the methods used to assess behaviours (e.g. self-report vs. device measure). For example, self-reported PA measures in previous studies indicated a higher occurrence of unfavourable health behaviours among girls compared to boys [37, 38]. Further prospective research conducted with more objective measures such as those used in the study are needed to provide a comprehensive understanding of the factors influencing different demographic groups.

A limited number of previous studies indicate a longitudinal association between persistent engagement in unhealthy behaviours and obesity-related measures [39–41]. For example, a study conducted in Germany explored the longitudinal association between patterns of self-reported health-related behaviours and changes in weight status and self-rated health among adolescents. It revealed that adolescents belonging to a poor health-related behaviour pattern characterised by low activity levels, high media use, and poor diet quality experienced the most

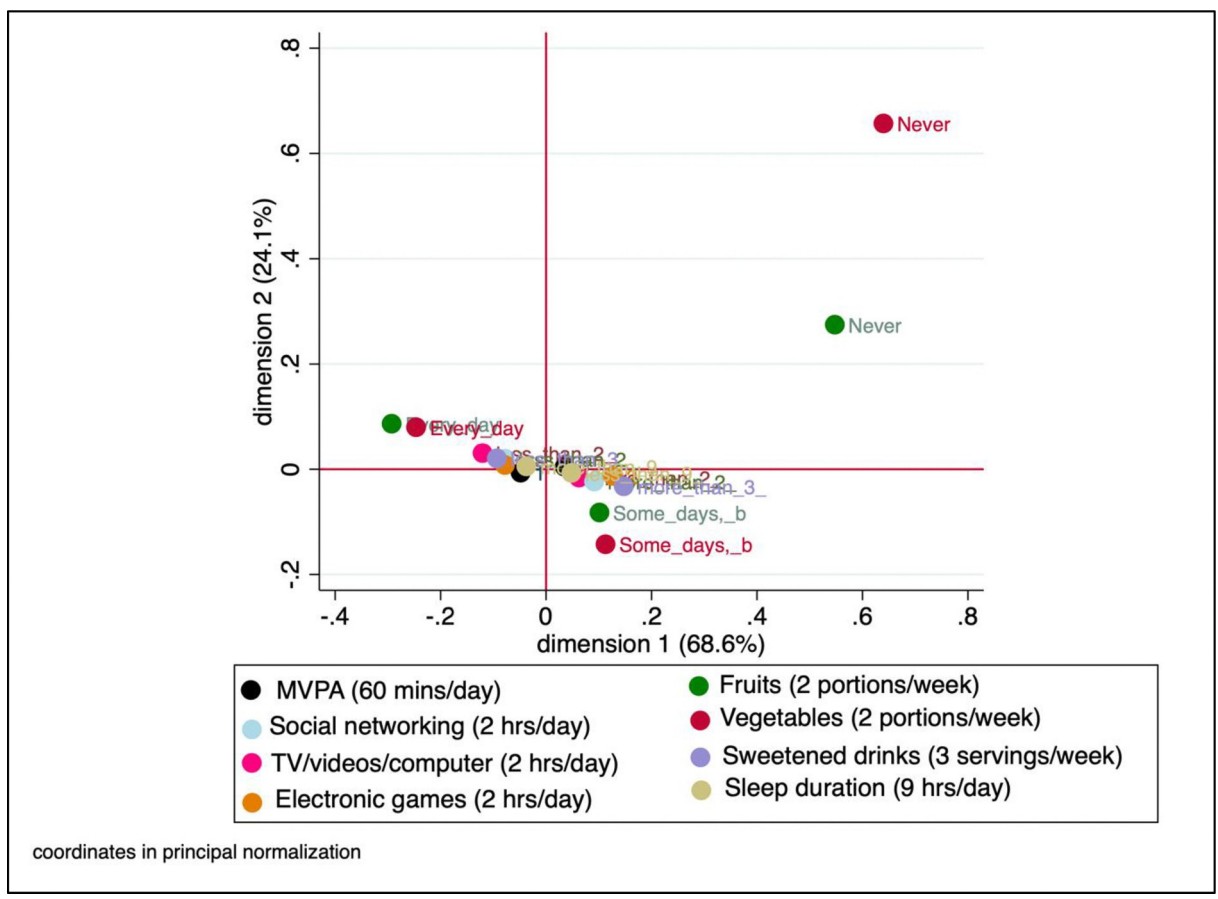

**Fig 1. Joint correspondence analysis plot of health behaviours variables.**

significant increase in overweight prevalence over a 6-year period. While the immediate impact of clusters of health behaviours on change in obesity related measures between 14 and 17 may not be apparent in the present study, the prevalence of these unhealthy behaviours underscores the need for proactive intervention and preventive measures to mitigate their potential long-term impact on health outcomes [15]. Addressing these behaviours early on is crucial to fostering healthier lifestyles and averting adverse health consequences in the future.

The present study revealed an unexpected trend where adolescent boys in the mixed and unhealthy clusters at baseline had a smaller change in BMI z-score between ages 14–17 compared to those in the healthy cluster. Although this finding diverges from conventional expectations, we posit that it is partially explained by boys with unhealthy behaviours having a higher BMI z-score (0.64 (SD = 1.17) at age 14 compared with the other cluster (healthy (0.42 (SD = 1.06)) and mixed (0.57 (SD = 1.20)), consequently experiencing less subsequent gain than their counterparts with healthier behaviours. Despite the slower gain in BMI z-scores in these clusters they still have higher mean BMI z-scores at age 17 than the other healthy cluster.

One crucial aspect that may require further emphasis is the substantial proportion of clusters categorised as overweight or obese at age 17. This observation sheds light on the significant prevalence of overweight and obesity within the studied population during late adolescence. Such a high prevalence underscores the pressing need for targeted interventions and preventive measures aimed at addressing obesity-related issues among adolescents. Understanding the factors within these clusters contributing to the persistence or development of overweight

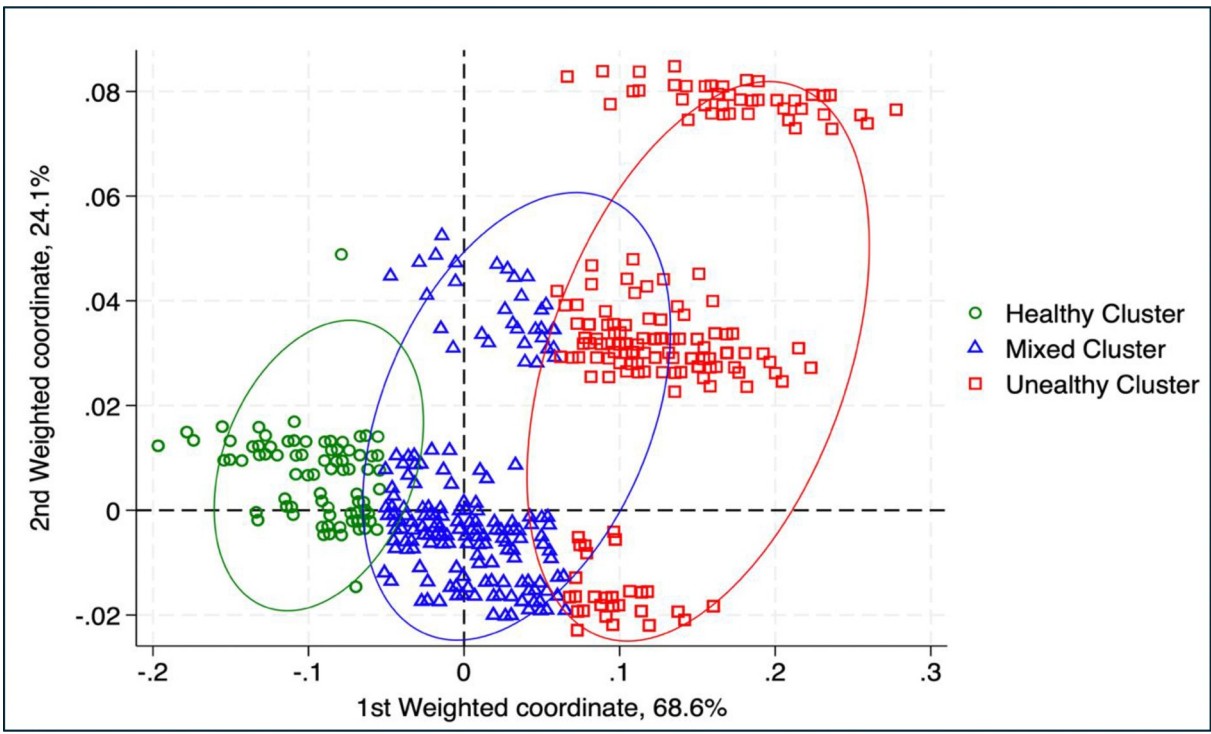

**Fig 2. K-means cluster analysis at age 14.**

and obesity in this age group is important for effective public health strategies and interventions to promote healthy weight management and mitigate associated health risks.

The finding that adolescent boys in the mixed and unhealthy clusters at baseline showed less change in BMI z-score over time may be attributed to the 'regression to the mean' (RTM) phenomenon [42]. The principle of RTM predicts greater mean weight change will occur in those participants with more extreme baseline body weight, leading to an inverse association between baseline body weight and expected weight change even without any intervention. This RTM confounds the interpretation of weight change, especially in those subjects whose baseline weight differs most from the population mean. In our study, this suggests that the initially lower prevalence of obesity in the 'healthy' cluster may have represented an outlier. Over time, the BMI values are converging towards the average, contributing to the unexpected longitudinal BMI change in this cluster. It's important to note the absence of this pattern when using the objective percent body fat measure, highlighting the challenges associated with using BMI for assessing obesity in adolescents due to its limitations in accurately capturing variations in body composition and the potential for misclassifying young people's weight status [43].

These findings can only be partially compared to a few studies that explored longitudinal associations between health behaviour clusters and adiposity change over time due to differences in the variables included across studies [39–41]. Notably, in contrast to the present study, a recent longitudinal study of 1634 Spanish adolescents aged 8–18 years found that participants belonging to the healthy behaviour cluster, characterised by low screen and sedentary time and high MVPA and adherence to the Mediterranean diet, showed significantly lower percent body fat two years later compared with the other profiles [16]. Another study of 1642 German adolescents showed that health behaviour clusters of low activity level with high

**Table 2. Demographic and behavioural characteristics of behavioural clusters at age 14 years split by gender.**

| Cluster | Cluster 1 Healthy Cluster N = 925 (30.18%) | Cluster 2 Mixed Cluster N = 1346 (43.92%) | Cluster 3 Unhealthy Cluster N = 794 (25.91%) | P value |
|---|---|---|---|---|
| **Gender (n (%))** | | | | 0.001 |
| Boys | 408 (44.11%) | 628 (46.66%) | 421 (53.02%) | |
| Girls | 517 (55.89%) | 718 (53.34%) | 373 (46.98%) | |
| **Gender** Boys | | | | |
| **Ethnicity (n (%))** | | | | 0.048 |
| White | 361 (88.48%) | 523 (83.28%) | 351 (83.37%) | |
| Other | 47 (11.52%) | 105 (16.72%) | 70 (16.63%) | |
| **Puberty (n (%))** | | | | 0.441 |
| Pre/early puberty | 285 (70.02%) | 425 (67.78%) | 276 (65.87%) | |
| Post/late pubertal | 122 (29.98%) | 202 (32.22%) | 143 (34.13%) | |
| **OECD weekly family income (GBP, mean (SD))** | 524 (156) | 450 (172) | 403 (161) | <0.001 |
| **NVQ Highest Level (all sweeps, n (%))** | | | | <0.001 |
| Low | 117 (28.68%) | 257 (40.92%) | 206 (48.93%) | |
| High | 284 (69.61%) | 341 (54.30%) | 186 (44.18%) | |
| Other qualifications | 7 (1.72%) | 30 (4.78%) | 29 (6.89%) | |
| **Physical activity (n (%))** | | | | <0.001 |
| Not meeting the guidelines* | 155 (37.99%) | 318 (50.64%) | 256 (60.81%) | |
| Meeting the guidelines* | 253 (62.01%) | 310 (49.36%) | 165 (39.19%) | |
| **Screen time (hrs/weekday, n (%))** | | | | |
| **Watching TV / videos / computer** | | | | <0.001 |
| < 2 | 258 (63.24%) | 204 (32.48%) | 68 (16.15%) | |
| > 2 | 150 (36.76%) | 424 (67.52%) | 353 (83.85%) | |
| **Playing electronic games** | | | | <0.001 |
| < 2 | 301 (73.77%) | 224 (35.67%) | 54 (12.83%) | |
| > 2 | 107 (26.23%) | 404 (64.33%) | 367 (87.17%) | |
| **Social networking sites** | | | | <0.001 |
| < 2 | 357 (87.50%) | 444 (70.70%) | 170 (40.38%) | |
| > 2 | 51 (12.50%) | 184 (29.30%) | 251 (59.62%) | |
| **Sleep duration (hrs/day, n (%))** | | | | <0.001 |
| < 9 | 130 (31.86%) | 264 (42.04%) | 244 (57.96%) | |
| > 9 | 278 (68.14%) | 364 (57.96%) | 177 (42.04%) | |

*(Continued)*

**Table 2.** (Continued)

| Cluster | Cluster 1<br>Healthy Cluster<br>N = 925<br>(30.18%) | Cluster 2<br>Mixed Cluster<br>N = 1346<br>(43.92%) | Cluster 3<br>Unhealthy Cluster<br>N = 794<br>(25.91%) | *P* value |
|---|---|---|---|---|
| **Dietary intake** | | | | |
| **Sweetened beverages (frequency/week, n (%))** | | | | <0.001 |
| < 3 times a week | 342<br>(83.82%) | 398<br>(63.38%) | 88<br>(20.90%) | |
| > 3 times a week | 66<br>(16.18%) | 230<br>(36.62%) | 333<br>(79.10%) | |
| **At least 2 portions of fruit per day (n (%))** | | | | <0.001 |
| Never / some days | 100<br>(24.51%) | 455<br>(72.45%) | 407<br>(96.67%) | |
| Every day | 308<br>(75.49%) | 173<br>(27.55%) | 14<br>(3.33%) | |
| **At least 2 portions of vegetables per day (n (%))** | | | | <0.001 |
| Never / some days | 64<br>(15.69%) | 420<br>(66.88%) | 395<br>(93.82%) | |
| Every day | 344<br>(84.31%) | 208<br>(33.12%) | 26<br>(6.18%) | |
| **Indicators of adiposity at age 14<br>(mean (SD))** | | | | |
| Body fat % at 14 | 14.91 (6.13) | 16.70 (8.26) | 16.95 (7.79) | <0.001 |
| BMI z-score at 14 | 0.42 (1.06) | 0.57 (1.20) | 0.64 (1.17) | 0.0002 |
| **BMI category 14 (n (%))** | | | | 0.032 |
| Normal/Under-weight | 336<br>(82.35%) | 483<br>(76.91%) | 319<br>(75.77%) | |
| Overweight | 61<br>(14.95%) | 104<br>(16.56%) | 73<br>(17.34%) | |
| Obese | 11<br>(2.70%) | 41<br>(6.53%) | 29<br>(6.89%) | |
| **Indicators of adiposity at age 17<br>(mean (SD))** | | | | |
| Body fat % at 17 | 14.66 (6.72) | 15.84 (8.59) | 16.17 (7.91) | <0.001 |
| BMI z-score at 17 | 1.16 (0.92) | 1.21 (1.12) | 1.26 (1.08) | 0.0027 |
| **BMI category 17 (n (%))** | | | | 0.001 |
| Normal/Under-weight | 322<br>(78.92%) | 466<br>(74.20%) | 302<br>(71.73%) | |
| Overweight | 71<br>(17.40%) | 98<br>(15.61%) | 75<br>(17.81%) | |
| Obese | 15<br>(3.68%) | 64<br>(10.19%) | 44<br>(10.45%) | |
| **Gender**<br>Girls | | | | |
| | **Cluster 1<br>Healthy Cluster** | **Cluster 2<br>Mixed Cluster** | **Cluster 3<br>Unhealthy Cluster** | *P* value |
| **Ethnicity (n (%))** | | | | 0.091 |
| White | 452<br>(87.43%) | 601<br>(83.70%) | 308<br>(82.57%) | |
| Other | 65<br>(12.57%) | 117<br>(16.30%) | 65<br>(17.43%) | |
| **Puberty (n (%))** | | | | <0.001 |
| Pre/early puberty | 76<br>(14.76%) | 57<br>(7.98%) | 34<br>(9.14%) | |

*(Continued)*

**Table 2.** (Continued)

| Cluster | Cluster 1 Healthy Cluster N = 925 (30.18%) | Cluster 2 Mixed Cluster N = 1346 (43.92%) | Cluster 3 Unhealthy Cluster N = 794 (25.91%) | P value |
|---|---|---|---|---|
| Post/late pubertal | 439 (85.24%) | 657 (92.02%) | 338 (90.86%) | |
| **OECD weekly family income (GBP, mean (SD))** | 508 (173) | 434 (171) | 384 (167) | <0.001 |
| **NVQ Highest Level (all sweeps, n (%))** | | | | <0.001 |
| Low | 165 (31.91%) | 323 (44.99%) | 187 (50.13%) | |
| High | 338 (65.38%) | 351 (48.89%) | 149 (39.95%) | |
| Other qualifications | 14 (2.71%) | 44 (6.13%) | 37 (9.92%) | |
| **Physical activity (n (%))** | | | | <0.001 |
| Not meeting the guidelines* | 278 (53.77%) | 490 (68.25%) | 273 (73.19%) | |
| Meeting the guidelines* | 239 (46.23%) | 228 (31.75%) | 100 (26.81%) | |
| **Screen time (hrs/weekday, n (%))** | | | | |
| **Watching TV / videos / computer** | | | | <0.001 |
| < 2 | 274 (53.00%) | 201 (27.99%) | 41 (10.99%) | |
| > 2 | 243 (47.00%) | 517 (72.01%) | 332 (89.01%) | |
| **Playing electronic games** | | | | <0.001 |
| < 2 | 494 (95.55%) | 604 (84.12%) | 220 (58.98%) | |
| > 2 | 23 (4.45%) | 114 (15.88%) | 153 (41.02%) | |
| **Social networking sites** | | | | <0.001 |
| < 2 | 352 (68.09%) | 279 (38.86%) | 52 (13.94%) | |
| > 2 | 165 (31.91%) | 439 (61.14%) | 321 (86.06%) | |
| **Sleep duration (hrs/day, n (%))** | | | | <0.001 |
| < 9 | 185 (35.78%) | 317 (44.15%) | 203 (54.42%) | |
| > 9 | 332 (64.22%) | 401 (55.85%) | 170 (45.58%) | |
| **Dietary intake** | | | | |
| **Sweetened beverages (frequency/week, n (%))** | | | | <0.001 |
| < 3 times a week | 462 (89.36%) | 496 (69.08%) | 91 (24.40%) | |
| > 3 times a week | 55 (10.64%) | 222 (30.92%) | 282 (75.60%) | |
| **At least 2 portions of fruit per day (n (%))** | | | | <0.001 |
| Never / some days | 114 (22.05%) | 589 (82.03%) | 367 (98.39%) | |
| Every day | 403 (77.95%) | 129 (17.97%) | 6 (1.61%) | |
| **At least 2 portions of vegetables per day (n (%))** | | | | <0.001 |
| Never / some days | 65 (12.57%) | 513 (71.45%) | 356 (95.44%) | |

(*Continued*)

**Table 2.** (Continued)

| Cluster | Cluster 1 Healthy Cluster N = 925 (30.18%) | Cluster 2 Mixed Cluster N = 1346 (43.92%) | Cluster 3 Unhealthy Cluster N = 794 (25.91%) | P value |
|---|---|---|---|---|
| Every day | 452 (87.43%) | 205 (28.55%) | 17 (4.56%) | |
| **Indicators of adiposity at age 14 (mean (SD))** | | | | |
| Body fat % at 14 | 26.08 (6.82) | 26.52 (6.50) | 27.91 (7.06) | <0.001 |
| BMI z-score at 14 | 0.58 (1.12) | 0.62 (1.11) | 0.80 (1.20) | 0.0002 |
| **BMI category 14 (n (%))** | | | | 0.046 |
| Normal/Under-weight | 398 (76.98%) | 559 (77.86%) | 260 (69.71%) | |
| Overweight | 88 (17.02%) | 118 (16.43%) | 82 (21.98%) | |
| Obese | 31 (6.00%) | 41 (5.71%) | 31 (8.31%) | |
| **Indicators of adiposity at age 17 (mean (SD))** | | | | |
| Body fat % at 17 | 27.47 (7.62) | 27.69 (7.92) | 29.03 (8.87) | <0.001 |
| BMI z-score at 17 | 1.05 (1.04) | 1.09 (1.07) | 1.24 (1.16) | 0.0027 |
| **BMI category 17 (n (%))** | | | | 0.017 |
| Normal/Under-weight | 375 (72.53%) | 518 (72.14%) | 236 (63.27%) | |
| Overweight | 98 (18.96%) | 129 (17.97%) | 89 (23.86%) | |
| Obese | 44 (8.51%) | 71 (9.89%) | 48 (12.87%) | |

Abbreviations: OECD, organisation for economic co-operation and development; GBP, great british pound; SD, standard deviation; NVQ, national vocational qualification; hrs, hours; BMI, body mass index. *>60 minutes of moderate to vigorous physical activity per day.

media use and low diet quality had the strongest increase in prevalence of overweight over a period of six years [41]. Similarly, Gubbels et al also revealed a longitudinal positive association between a 'sedentary (TV viewing)/snacking' pattern at five years of age and BMI Z-score at 7–8 years [40]. The inconsistency in the evidence with the present study can be partly linked to the utilisation of different methods for assessing health behaviours [44], as well as the different age group focuses of the studies, and different time frames of follow up. Moreover, many large-scale population-based studies rely on respondent self-reports to measure physical activity, which are susceptible to both recall and social desirability bias [45]. Also, as the questions assessing SSB consumption do not measure portion size, the under-reporting of SSB intake with the over-reporting of fruits and vegetables by adolescents with a higher BMI might have occurred, because overweight/obese adolescents are inclined to under-report their unhealthy dietary intake more often than adolescents with normal weight [46].

## Strengths and limitations

This present study has several methodological strengths. First, it is based on data from a large, representative cohort of adolescents with a three year follow up. Second, the longitudinal study design and adjustment for baseline adiposity lowers the risk of reverse causation [47]. Third, percent body fat, via calibrated Tanita BF-522W scales, was utilised which is a more direct

**Table 3. Association between behaviour clusters at age 14 and adiposity markers at age 17, unadjusted models.**

| | BMI z-score $\Delta$ | | P value | Weight status $\Delta$ | | P value | % BF $\Delta$ | | P value |
|---|---|---|---|---|---|---|---|---|---|
| | β | 95% CI | | Odds ratio | 95% CI | | β | 95% CI | |
| Boys | | | | | | | | | |
| Cluster 1 Healthy (Reference) | | | | | | | | | |
| Cluster 2 Mixed | 0.051 | (- 0.081)—(0.182) | 0.448 | 1.302 | 0.967–1.753 | 0.083 | 1.177 | 0.190–2.163 | **0.019** |
| Cluster 3 Unhealthy | 0.103 | (- 0.040)—(0.247) | 0.159 | 1.475 | 1.073–2.029 | **0.017** | 1.512 | 0.434–2.590 | **0.006** |
| Girls | | | | | | | | | |
| Cluster 1 Healthy (Reference) | | | | | | | | | |
| Cluster 2 Mixed | 0.040 | (- 0.083)—(0.162) | 0.530 | 1.020 | 0.792–1.313 | 0.880 | 0.221 | (- 0.691)—(0.133) | 0.635 |
| Cluster 3 Unhealthy | 0.191 | (0.046)—(0.335) | **0.010** | 1.533 | 1.152–2.040 | **0.003** | 1.556 | 0.483–2.630 | **0.005** |

Abbreviations: BMI, body mass index; %BF, percent body fat; 95%CI: 95% confidence interval.

measure of adiposity compared to BMI [48]. Lastly, device-based measures of MVPA reduced the bias associated with self-report [49]. It is important to consider the following limitations when interpreting the results. We acknowledge that a substantial number of participants were excluded from the analysis due to missing accelerometer data, consistent with previous research using this methodology [50]. The participants excluded from the analysis differed in terms of ethnicity and socioeconomic status compared to those included. The analytical sample was representative of the home countries, however the findings may have limited the

**Table 4. Association between behaviour clusters at age 14 and adiposity markers at age 17, adjusted models.**

| | BMI z-score $\Delta$ | | P value | Weight status $\Delta$ | | P value | % BF $\Delta$ | | P value |
|---|---|---|---|---|---|---|---|---|---|
| | β | 95% CI | | Odds ratio | 95% CI | | β | 95% CI | |
| Boys | | | | | | | | | |
| Cluster 1 Healthy (Reference) | | | | | | | | | |
| Cluster 2 Mixed | - 0.08 | (- 0.146)–(-0.005) | **0.036** | 0.96 | 0.644–1.428 | 0.835 | - 0.33 | (- 1.044)—(0.386) | 0.367 |
| Cluster 3 Unhealthy | - 0.08 | (- 0.158)–(- 0.001) | **0.048** | 1.08 | 0.700–1.671 | 0.725 | - 0.31 | (- 1.107)—(0.491) | 0.450 |
| Girls | | | | | | | | | |
| Cluster 1 Healthy (Reference) | | | | | | | | | |
| Cluster 2 Mixed | 0.001 | (- 0.063)—(0.065) | 0.976 | 0.96 | 0.673–1.369 | 0.820 | -0.289 | (- 0.874)—(0.296) | 0.333 |
| Cluster 3 Unhealthy | -0.007 | (- 0.083)—(0.069) | 0.853 | 1.20 | 0.792–1.810 | 0.393 | -0.287 | (- 0.988)—(0.414) | 0.422 |

Abbreviations: BMI, body mass index; %BF, percent body fat; 95%CI: 95% confidence interval. Model was adjusted for maternal education, weekly family income, ethnicity, birthweight and puberty, baseline BMI z-score, baseline weight status, baseline percent body fat, respectively.

generalisability to the broader UK population due to the reduced sample size utilised. Moreover, relying on just two valid days of accelerometer data, including one weekend day and one weekday, to measure MVPA may not adequately reflect an individual's regular physical activity patterns [51]. However, it is worth noting that some existing data do suggest that a measurement period of two days can provide reliable insights into overall physical activity levels [52]. Physical activity data was collected across a 14-month period and thus seasonal variation could have influenced physical activity levels [53]. The validity and reliability of the questionnaires used to assess screen time, food habits and sleep were not independently tested, as they were developed by the cohort management team, which may affect the accuracy of the findings. Additionally, screen time habits in young adolescents may have changed since they were initially measured in 2015 and 2016 in the MCS [31]. Also, relying on fruits, vegetables and SSB to measure dietary behaviours and excluding other important diet components is a limitation in relation to understanding overall energy intakes and diversity of diet. It is important also to emphasise that even young people within the healthy cluster did not necessarily adhere to recommended dietary guidelines (i.e. 5 portions of fruit and vegetables daily). There could also have been a measurement error with the dietary behaviours data because the self-reported questionnaire used are prone to biases, such as attention and recall bias [46]. Furthermore, the questions utilised in this study asked about frequency of consuming SSBs but not amount consumed, which limits our ability to conclude on the volume of SSB consumed. Moreover, while BMI is widely used, it may not accurately reflect body composition in adolescents. Future research should consider the use of the ponderal index which might be more suitable where there is variability in height and body composition could affect the accuracy of BMI. Finally, we are not able to examine whether clusters tracked between the ages of 14 and 17 due to limitations with the available MCS data.

## Conclusion

To our knowledge, this is the first longitudinal study among older adolescents in the UK to examine the associations between clusters of health behaviours and change in indicators of adiposity over time. Adolescents in the healthy cluster had lower BMI z-scores and percent body fat at age 14 years than those in the unhealthy cluster, and they maintained this lower level at age 17. Our findings show that for most of the indicators of adiposity measured across both boys and girls that there were no significant differences in change in these outcomes between the health behaviour clusters between 14–17 years. This was with the exception of BMI z-score where there was a marginally statistically significant finding with a small effect size showing the unhealthy and mixed clusters to have had less change in BMI z-scores during this period from a relatively higher start in BMI z-scores compared to the healthy cluster. To support future research, there is a clear need to establish prospective research conducted with better objective measures of diet, sleep, and screen time. Also, as adolescents' health behaviours are complex and undergo changes with age [54], further research is necessary to comprehensively understand and analyse these evolving patterns. Additionally, examining the stability of cluster membership over time is needed.

## Supporting information

**S1 Table. Descriptive analysis of demographic characteristics comparing the included and excluded populations.** Abbreviations: OECD, organisation for economic co-operation and development; NVQ, national vocational qualification.
(DOCX)

**S2 Table. Demographic and behavioural characteristics of behavioural clusters at age 14 years.** Abbreviations: OECD, organisation for economic co-operation and development; NVQ, national vocational qualification; hrs, hours; BMI, body mass index. *>60 minutes of moderate to vigorous physical activity per day.
(DOCX)

**S1 Fig. Sample selection flow-chart.**
(DOCX)

## Author Contributions

**Formal analysis:** Noura Alosaimi.

**Supervision:** Lauren B. Sherar, Paula Griffiths, Natalie Pearson.

**Writing – original draft:** Noura Alosaimi.

**Writing – review & editing:** Lauren B. Sherar, Paula Griffiths, Mark Hamer, Natalie Pearson.

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
