## [Decision Letter · Decision Letter 0]

1 Nov 2024

PONE-D-24-34337Clusters of Diet, Physical Activity, Screen-Time, and Sleep Among Adolescents and Associations with 3-Year Change in Indicators of AdiposityPLOS ONE

Dear Dr. Alosaimi,

Thank you for submitting your manuscript to PLOS ONE. After careful consideration, we feel that it has merit but does not fully meet PLOS ONE’s publication criteria as it currently stands. Therefore, we invite you to submit a revised version of the manuscript that addresses the points raised during the review process.

We look forward to receiving your revised manuscript.

Kind regards,

Mohamed Ahmed Said, Ph.D.

Academic Editor

PLOS ONE

Additional Editor Comments:

The editor has two remakes:

1. Initially, the MCS study involved a representative sample of 19,243 babies from the four constituent countries of the UK. However, the present study includes just over half of these participants, raising the question: how representative is the new sample? It’s possible that the participants are primarily from only two or three of these countries.

2. Figure 1 looks cluttered, with overlapping elements and almost similar colors used for some variables (e.g., sleep duration and TV; MVPA and sugary drinks). For greater clarity, could you divide it into two sub-figures? The suggested sub-figures could be: (a) physical activity and sedentary behaviors, and (b) eating behaviors.

Reviewers' comments:

Reviewer's Responses to Questions

**Comments to the Author**

1. Is the manuscript technically sound, and do the data support the conclusions?

Reviewer #1: Yes

Reviewer #2: Yes

2. Has the statistical analysis been performed appropriately and rigorously? 

Reviewer #1: Yes

Reviewer #2: Yes

3. Have the authors made all data underlying the findings in their manuscript fully available?

Reviewer #1: Yes

Reviewer #2: Yes

4. Is the manuscript presented in an intelligible fashion and written in standard English?

Reviewer #1: Yes

Reviewer #2: Yes

5. Review Comments to the Author

Reviewer #1: General comments

The authors have clearly stated that the study aimed to examine the clustering of screen time, physical activity, dietary behaviours and sleep, and the associations with 3-year changes in indicators of adiposity. The paper is well-written and easy to follow. The findings contribute to the growing body of evidence supporting the importance of health-related behaviours among adolescents with excess body weight or adiposity. In light of this approach, this work can inform future endeavors in a similar research area. However, a few comments and suggestions have been highlighted in the specific comments section below, which must be addressed before this work can be accepted for publication.

Specific comments

INTRODUCTION

- Add a statement about the popularity of physical activity programs for youth with overweight/obesity worldwide (1).

Suggested references:

1. DOI: 10.1249/FIT.0000000000000933

METHODS

- Preliminary power analysis results should be added to justify the selected sample size.

RESULTS

- Add effect sizes to the text and/or tables where significant differences were found.

DISCUSSION

- Compare the present results with other relevant articles in brief, aiming to help readers get familiar with the current evidence in this particular research area with a focus on the role of health-related behaviours among youth people (2-4).

Suggested references:

2. DOI: 10.3934/publichealth.2024040

3. DOI: 10.53555/ks.v12i2.2845

4. DOI: 10.1155/2024/5095049

Reviewer #2: Dear Authors,

Addressing this research topic is particularly valuable as it provides insight into how the interplay of key lifestyle factors influences the development of obesity during adolescence. This approach enables the identification of at-risk groups and supports the development of comprehensive health interventions. Additionally, this study offers valuable information on preventing adolescent obesity, the effects of which often persist into adulthood.

The study description indicates that participants were asked about their sleep time. It appears, however, that data from the GENEActiv triaxial accelerometer could be used to determine sleep time as well. Thanks to their high sensitivity and ability to track movement in three axes, GENEActiv accelerometers are commonly used in sleep studies to monitor physical activity and identify periods of rest and movement. Based on this data, it is possible to estimate sleep phases and quality.

Using Body Mass Index (BMI) to assess obesity in 14- and 17-year-olds is common practice but has some limitations. BMI is a simple index based on body weight and height, though its accuracy in children and adolescents may be reduced because it does not account for developmental differences and body proportions that vary during adolescence. For adolescents, it should always be interpreted according to age- and sex-specific centile grids, as the reference ranges for BMI differ between children/adolescents and adults.

BMI remains the most widely used tool due to its well-established role in research and the availability of centile grids. However, the Ponderal Index (PI) may be more suitable in cases where significant variability in height and body composition could affect BMI accuracy. In practice, using both indices concurrently and tracking their changes over time can provide a more comprehensive picture of adolescent health.

Congratulations on your choice of topic and the successful completion of this study!

Detailed suggestions are provided in the comments.

6. PLOS authors have the option to publish the peer review history of their article (what does this mean?). If published, this will include your full peer review and any attached files.

Reviewer #1: No

Reviewer #2: **Yes: **Joanna Baj-Korpak

---

## [Author Response · Author response to Decision Letter 0]

13 Nov 2024

Reviewer Point # Reviewer Comments Response

1 1. Initially, the MCS study involved a representative sample of 19,243 babies from the four constituent countries of the UK. However, the present study includes just over half of these participants, raising the question: how representative is the new sample? It’s possible that the participants are primarily from only two or three of these countries.

Thank you taking the time to review this manuscript and for the helpful comments. We have responded to each comment in turn below.

Although we set specific inclusion criteria eg., requiring participants have valid activity monitor data, and complete data on all health behaviours and outcomes of interest, which reduced the sample size, the sample remained representative. We have provided a sentence in the results with the data to show the numbers from each of the constituent countries. Page 12, lines 279-281

“Participants included in the analyses represented all UK countries with 30% of participants from England, 37% from Wales, 42% from Northern Ireland, and 44% from Scotland.”

 2 Figure 1 looks cluttered, with overlapping elements and almost similar colors used for some variables (e.g., sleep duration and TV; MVPA and sugary drinks). For greater clarity, could you divide it into two sub-figures? The suggested sub-figures could be: (a) physical activity and sedentary behaviors, and (b) eating behaviors.

Thank you for your feedback on Figure 1. This figure illustrates the dimensions of health behaviours derived from the correspondence analysis, which requires all health behaviours to be included together to capture the relationships among them effectively. 

Correspondence analysis is a multivariate statistical technique that allows us to visualise categorical data by representing variables and observations in a two-dimensional space. By analysing the associations among all health behaviours collectively, we gain insights into their interrelationships.

While we understand the concern regarding clutter and overlapping elements, separating the variables into sub-figures according to health behaviours would compromise the integrity of the analysis and would not accurately reflect the aims of this study which is to examine the clustering of the four health behaviours.

2 1 

The authors have clearly stated that the study aimed to examine the clustering of screen time, physical activity, dietary behaviours and sleep, and the associations with 3-year changes in indicators of adiposity. The paper is well-written and easy to follow. The findings contribute to the growing body of evidence supporting the importance of health-related behaviours among adolescents with excess body weight or adiposity. In light of this approach, this work can inform future endeavors in a similar research area. However, a few comments and suggestions have been highlighted in the specific comments section below, which must be addressed before this work can be accepted for publication.

 Thank you for taking the time to read this manuscript and for your insightful comments. We have responded to each of your comments below.

 2 

INTRODUCTION

Add a statement about the popularity of physical activity programs for youth with overweight/obesity worldwide (1).

Suggested references:

1. DOI: 10.1249/FIT.0000000000000933

We could see merit in adding such a statement/reference if the focus of the introduction/manuscript was on physical activity. However, this study has examined the clustering of four health behaviours and feel that such a statement does not fit in the introduction as the focus would be drawn to physical activity only, and not the clustering of multiple behaviours.

 3 METHODS

Preliminary power analysis results should be added to justify the selected sample size. 

Thank you for your suggestion regarding the inclusion of preliminary power analysis results to justify the selected sample size. Given that this analysis is of secondary data, the original sample size was determined by the available data from the Millennium Cohort Study. As such, the study is based on a specific subset of participants who met the criteria of consenting to wear activity monitors and having complete data on health behaviours and outcomes.

As described in the manuscript (page 10, lines 240-243), we conducted a sensitivity analysis comparing participants with complete data to those with missing data on key health outcomes and found a significant difference in one outcome. These findings supported our decision to use a complete-case dataset, ensuring the accuracy and reliability of our analyses. 

“A sensitivity analysis showed a significant difference in one key health outcome between participants with complete and missing data, supporting the use of a complete-case dataset for accuracy and reliability. Therefore, only participants with complete data for all variables of interest at ages 14 and 17 were included.”

 4 RESULTS

Add effect sizes to the text and/or tables where significant differences were found. 

Given the volume of data and information in the tables, we have provided the readers with the beta-coefficient, confidence intervals and p-value which is standard practice for this type of analysis. 

 5 

DISCUSSION

Compare the present results with other relevant articles in brief, aiming to help readers get familiar with the current evidence in this particular research area with a focus on the role of health-related behaviours among youth people (2-4).

Suggested references:

2. DOI: 10.3934/publichealth.2024040

3. DOI: 10.53555/ks.v12i2.2845

4. DOI: 10.1155/2024/5095049

We appreciate your comment on comparing our results with other relevant articles, and thank you for the suggested references. However, we feel that the discussion section already compares our findings with the most up to date relevant articles. The references you have provided are for cross-sectional data that do not include the specific health behaviours that we have included in these analyses so we don’t feel that they would be relevant in the discussion section.

3 1 

Addressing this research topic is particularly valuable as it provides insight into how the interplay of key lifestyle factors influences the development of obesity during adolescence. This approach enables the identification of at-risk groups and supports the development of comprehensive health interventions. Additionally, this study offers valuable information on preventing adolescent obesity, the effects of which often persist into adulthood.

 Thank you for taking the time to review this manuscript and for the insightful comments. We have addressed each comment below.

 2 

The study description indicates that participants were asked about their sleep time. It appears, however, that data from the GENEActiv triaxial accelerometer could be used to determine sleep time as well. Thanks to their high sensitivity and ability to track movement in three axes, GENEActiv accelerometers are commonly used in sleep studies to monitor physical activity and identify periods of rest and movement. Based on this data, it is possible to estimate sleep phases and quality.

Thank you for highlighting the capabilities of the GENEActiv accelerometer. As this study uses secondary data, the use of the accelerometer was set a-priori and not by the authors of this manuscript. In the original protocol for the MCS study, the GENEActive was used to assess physical activity only. We were able to utilise the data available for the MCS cohort and thus utilised the self-reported sleep duration measures.

 3 

Using Body Mass Index (BMI) to assess obesity in 14- and 17-year-olds is common practice but has some limitations. BMI is a simple index based on body weight and height, though its accuracy in children and adolescents may be reduced because it does not account for developmental differences and body proportions that vary during adolescence. For adolescents, it should always be interpreted according to age- and sex-specific centile grids, as the reference ranges for BMI differ between children/adolescents and adults.

BMI remains the most widely used tool due to its well-established role in research and the availability of centile grids. However, the Ponderal Index (PI) may be more suitable in cases where significant variability in height and body composition could affect BMI accuracy. In practice, using both indices concurrently and tracking their changes over time can provide a more comprehensive picture of adolescent health.

Thank you for your insightful comment on the use of BMI in adolescents. As this study is based on secondary data, the BMI values were already calculated and provided within the dataset. I acknowledge that while BMI has limitations in accurately reflecting body composition in adolescents, it remains the most widely used and accessible measure, especially in large-scale datasets. 

We have noted this limitation in the manuscript on page 26, lines 510-513.

“Moreover, while BMI is widely used, it may not accurately reflect body composition in adolescents. Future research should consider the use of the ponderal index which might be more suitable where there is variability in height and body composition could affect the accuracy of BMI.”

Comment within the Manuscript 1 

Physical activity 

(Pages 6-7)

When was the PA measured? Does the time of year make a difference to the physical activity undertaken? Was this not a limitation of the study?

Thank you for raising this question. Physical activity data was measured between January 2015 and March 2016. We have added the time period for data collection to the manuscript on page 7, lines 143-144.

In this study, accelerometer data was collected over a 14-month period, so seasonal variation may have influenced physical activity levels. This has been added as a limitation in the discussion on page 25, lines 497–498.

“Physical activity data was collected across a 14-month period and thus seasonal variation could have influenced physical activity levels.”

 2 

Screen time

(page 7)

How did they report? Please explain 

Screen time was self-reported using a likert scale for each of the screen-based behaviours. We have edited this section to provide more clarity (page 7, lines 154-158)

“To examine time spent on screen-based behaviours at age 14 years (MCS6 survey), adolescents self-reported the number of hours per weekday spent (i) watching tv / videos / computers, (ii) playing electronic games, and (iii) on social networking sites. Adolescents responded to an 8-point Likert scale ranging from ‘none’ to ‘7 or more hours a day’. For the present analyses, each screen-time variable was categorised as < 2 hours and > 2 hours[27].”

 3 

Dietary behaviours

(pages 7-8)

In addition to frequency, was the amount of sweetened drinks consumed analysed? 

This seems relevant (if not analysed, I suggest including this in the limitations of the study)

This was a secondary analysis of an existing cohort; the amount of sweetened drinks was not measured in this study and we have acknowledged this as a limitation on page 26, lines 508-510.

 4 

Sleep 

(page 8)

Were the subjects asked about sleep? It appears that data from the GENEActiv triaxial accelerometer can be used to determine sleep time. Thanks to their high sensitivity and ability to track movement in three axes, GENEActiv accelerometers are often used in sleep studies to monitor physical activity and determine the user's moments of rest and movement. On this basis, estimating sleep phases and quality is possible. 

Yes, participants were asked about their sleep as documented on page 8, lines 175-186 in the methods section.

Thank you for highlighting the capabilities of the GENEActiv accelerometer. As this study uses secondary data, the use of the accelerometer was set a-priori and not by the authors of this manuscript. In the original protocol for the MCS study, the GENEActive was used to assess physical activity only. We were able to utilise the data available for the MCS cohort and thus utilised the self-reported sleep duration measures.

 5 

Adiposity

(pages 8-9)

Has the use of PI not been considered? This indicator may be better in situations where high variability in height and weight may affect the BMI result. 

Given that this study uses secondary data from MCS Wave 6, the analysis relied on BMI values that were already calculated and provided in the dataset. While PI may offer advantages in cases with high variability in height and weight, alternative measures were not available in the data. We have noted the importance of using PI in future research in the study's limitations on page 26, lines 510-513.

“Moreover, while BMI is widely used, it may not accurately reflect body composition in adolescents. Future research should consider the use of the ponderal index which might be more suitable where there is variability in height and body composition could affect the accuracy of BMI.”

 6 

Participant characteristics 

(pages 11-12)

This is a section of the paper dealing with an aspect of the research methodology, i.e. the study material (characteristics of the participants). In my opinion it is not part of the results. I suggest relocating as an earlier point in the manuscript.

The information in the participant characteristics section is data on the results – i.e., the proportion of the sample that were boys and girls for example. As per standard practice, these results are documented in a Table of results and described in the results section.

 7 

Table 1

(page 12)

Weekly family income

unit (currency)

The currency, GBP, has been added to the table for clarity.

 8 

Table 1

(page 13)

MVPA (mins/day) 

Is the value 52.25 an integer value? This is indicated by the title of the column.

The value represents the median minutes per day (IQR), and this has been edited in the table for clarity.

 9 

Table 1

(page 13)

Screen time (mins/weekday) 

See comment above.

what do the ranges in brackets mean?

The values in brackets represent the Interquartile Range (IQR) for screen time (in minutes per weekday), indicating the spread of the middle 50% of the data around the median.

This has been edited in the table for clarity.

 10 

Table 1

(page 13)

Body fat % at 14 

The total value? Or the mean (median) value?

The value is mean (SD) and this has been added to the table for clarity.

 11 

Page 23, line 440

“A second hypothesis to explain the finding that adolescent boys….”

Have hypotheses been formulated in the paper? 

No, we did not formulate hypotheses for these analyses as they were exploratory. We have reworded the sentence for clarity. Page 23, lines 440-442.

“The finding that adolescent boys in the mixed and unhealthy clusters at baseline showed less change in BMI z-score over time may be attributed to the 'regression to the mean' (RTM) phenomenon.”

 12 

Supporting information, S1 Fig. Sample selection flow-chart. 

(Page 31)

Figure is illegible - difficult to analyse (overlapping data) 

Thank you for the feedback. The text size in the flowchart has been increased to improve readability. However, I haven’t observed any overlapping boxes in this figure.

---

## [Decision Letter · Decision Letter 1]

25 Nov 2024

PONE-D-24-34337R1Clusters of Diet, Physical Activity, Screen-Time, and Sleep Among Adolescents and Associations with 3-Year Change in Indicators of AdiposityPLOS ONE

Dear Dr. Alosaimi,

Thank you for submitting your manuscript to PLOS ONE. After careful consideration, we feel that it has merit but does not fully meet PLOS ONE’s publication criteria as it currently stands. Therefore, we invite you to submit a revised version of the manuscript that addresses the points raised during the review process.

Despite the authors' best efforts, some issues persist and necessitate additional attention before accepting the work. 

Please submit your revised manuscript within Jan 09 2025 11:59PM. If you will need more time than this to complete your revisions, please reply to this message or contact the journal office at plosone@plos.org. Please include the following items when submitting your revised manuscript:A rebuttal letter that responds to each point raised by the academic editor and reviewer(s). You should upload this letter as a separate file labeled 'Response to Reviewers'.A marked-up copy of your manuscript that highlights changes made to the original version. You should upload this as a separate file labeled 'Revised Manuscript with Track Changes'.An unmarked version of your revised paper without tracked changes. You should upload this as a separate file labeled 'Manuscript'.If applicable, we recommend that you deposit your laboratory protocols in protocols.io to enhance the reproducibility of your results. Protocols.io assigns your protocol its own identifier (DOI) so that it can be cited independently in the future. For instructions see: https://journals.plos.org/plosone/s/submission-guidelines#loc-laboratory-protocols. Additionally, PLOS ONE offers an option for publishing peer-reviewed Lab Protocol articles, which describe protocols hosted on protocols.io. Read more information on sharing protocols at https://plos.org/protocols?utm_medium=editorial-email&utm_source=authorletters&utm_campaign=protocols.

We look forward to receiving your revised manuscript.

Kind regards,

Mohamed Ahmed Said, Ph.D.

Academic Editor

PLOS ONE

Journal Requirements:

Additional Editor Comments:

Despite the authors' efforts, numerous topics remain unresolved and require additional attention.

1- The inceased reduction in sample size may impact the results' representativeness, generalizability, statistical power, and precision, potentially undermining the validity and trustworthiness of the study's conclusions. Presumably, the initial sample aimed to represent the diversity of the UK population, taking into account socioeconomic, regional, and demographic factors. A relative smaller sample size may limit the capacity to apply the findings to the total UK population, particularly if specific subgroups are underrepresented. Although the authors aimed to prove the sample's representativeness, they did not provide evidence of the representation of all significant subgroups. The investigation of potential confounding variables represents a step forward, but further work is required. To address this issue, the authors should investigate further or acknowledge it as a study limitation.

2- The validity and reliability of the questionnaires used to assess screen time and food habits should be developed.

3- In Figure, the data is stacked in a way that makes it difficult to differentiate between categories, as the colors are very similar.

Reviewers' comments:

Reviewer's Responses to Questions

**Comments to the Author**

1. If the authors have adequately addressed your comments raised in a previous round of review and you feel that this manuscript is now acceptable for publication, you may indicate that here to bypass the “Comments to the Author” section, enter your conflict of interest statement in the “Confidential to Editor” section, and submit your "Accept" recommendation.

Reviewer #1: (No Response)

Reviewer #2: All comments have been addressed

2. Is the manuscript technically sound, and do the data support the conclusions?

Reviewer #1: Partly

Reviewer #2: (No Response)

3. Has the statistical analysis been performed appropriately and rigorously? 

Reviewer #1: I Don't Know

Reviewer #2: (No Response)

4. Have the authors made all data underlying the findings in their manuscript fully available?

Reviewer #1: Yes

Reviewer #2: (No Response)

5. Is the manuscript presented in an intelligible fashion and written in standard English?

Reviewer #1: Yes

Reviewer #2: (No Response)

6. Review Comments to the Author

Reviewer #1: The present work does not fully meet the criteria to be accepted for publication in this journal. Particularly, the study has not been registered in an international database to promote credibility, clarity and transparency. Additionally, the manuscript was not revised as suggested in the first round of the peer review process.

Reviewer #2: (No Response)

7. PLOS authors have the option to publish the peer review history of their article (what does this mean?). If published, this will include your full peer review and any attached files.

Reviewer #1: No

Reviewer #2: **Yes: **Joanna Baj-Korpak

---

## [Author Response · Author response to Decision Letter 1]

5 Dec 2024

Editor

Point # Editor Comments Response

1 

The increased reduction in sample size may impact the results' representativeness, generalizability, statistical power, and precision, potentially undermining the validity and trustworthiness of the study's conclusions. Presumably, the initial sample aimed to represent the diversity of the UK population, taking into account socioeconomic, regional, and demographic factors. A relative smaller sample size may limit the capacity to apply the findings to the total UK population, particularly if specific subgroups are underrepresented. Although the authors aimed to prove the sample's representativeness, they did not provide evidence of the representation of all significant subgroups. The investigation of potential confounding variables represents a step forward, but further work is required. To address this issue, the authors should investigate further or acknowledge it as a study limitation.

Thank you for raising this question. 

We have added the below sentence as a limitation in the discussion on page 25, lines 427-429.

“The analytical sample was representative of the home countries, however the findings may have limited the generalisability to the broader UK population due to the reduced sample size utilised.”

2 

The validity and reliability of the questionnaires used to assess screen time and food habits should be developed.

We have clarified in the methods section (page 6, lines 120-121) that the questionnaires used to assess screen time, food habits and sleep were developed by the cohort management team.

“This is a secondary data analysis, and all measures (i.e. questionnaires, protocols for device use) were developed by the original cohort team.”

Also, we have added the below sentence as a limitation in the discussion on page 25, lines 434-437.

“The validity and reliability of the questionnaires used to assess screen time, food habits and sleep were not independently tested, as they were developed by the cohort management team, which may affect the accuracy of the findings.”

3 

In Figure, the data is stacked in a way that makes it difficult to differentiate between categories, as the colors are very similar.

Thank you for raising this concern regarding the clarity of the figure and the difficulty in differentiating between categories due to the similar colours.

We have edited the colours of all variables to make them more distinct.

---

## [Editor Report · Decision Letter 2]

8 Dec 2024

Clusters of Diet, Physical Activity, Screen-Time, and Sleep Among Adolescents and Associations with 3-Year Change in Indicators of Adiposity

PONE-D-24-34337R2

Dear Dr. Noura Alosaimi,

We’re pleased to inform you that your manuscript has been judged scientifically suitable for publication and will be formally accepted for publication once it meets all outstanding technical requirements.

Kind regards,

Mohamed Ahmed Said, Ph.D.

Academic Editor

PLOS ONE
---

## [Editor Report · Acceptance letter]

11 Dec 2024

PONE-D-24-34337R2 

PLOS ONE

Dear Dr. Alosaimi, 

I'm pleased to inform you that your manuscript has been deemed suitable for publication in PLOS ONE. Congratulations! Your manuscript is now being handed over to our production team.

Kind regards, 

on behalf of

Dr. Mohamed Ahmed Said 

Academic Editor

PLOS ONE